# The Healthy Brain 9 (HB9): A new instrument to characterize subjective cognitive decline, and detect anosognosia in mild cognitive impairment

James E. Galvin *, Katherine C. Almonte, Andrea Buehler, Yolene M. Caicedo, Conor B. Galvin, Willman Jimenez, Mahesh S. Joshi, Nicole Mendez, Mary Lou A. Riccio, Marcia I. Walker, Michael J. Kleiman

Comprehensive Center for Brain Health, Department of Neurology, University of Miami Miller School of Medicine, Boca Raton, Florida, United States of America

* jeg200@miami.edu

## Abstract

### Objectives

Subjective cognitive decline (SCD) affects 10% of older adults and may be a risk factor for future mild cognitive impairment (MCI) and dementia. Some individuals with MCI have anosognosia, the denial or lack of awareness of their cognitive deficits. We developed and tested the Healthy Brain 9 (HB9), a self-reported assessment of cognitive performance and everyday functioning, in a diverse community-based cohort of older adults in South Florida.

### Design

Cross-sectional study.

### Setting

Community-based longitudinal study of brain health.

### Participants

A total of 344 participants (mean age of 68.5±9.3y, 70% were female, 62% with 16 or less years of education, 39% ethnoracial minorities) completed the study. The sample included 42% normal cognition, 27% SCD and 30% MCI. Within the MCI group, 62% demonstrated awareness of cognitive deficits and 38% had MCI with anosognosia.

### Measurements

The psychometric properties of the HB9 were examined and the performance of the HB9 was compared to Gold Standard comprehensive clinical-cognitive-functional-behavioral evaluations and biomarkers evaluations from the Healthy Brain Initiative at the University of Miami.

**Data availability statement:** All relevant data are available in the Open Science Framework repository: https://doi.org/10.17605/OSF.IO/T9Q8S.

**Funding:** This work was supported by the National Institutes of Health (R01AG071514, R01AG071514S1, R01NS101483, R01NS101483S1, RF1AG075901, and R56AG074889 to JEG).

**Competing interests:** Dr Galvin is the creator of the HB9 and the copyright is held by the University of Miami Miller School of Medicine. Dr Galvin is Chief Scientific Officer for Cognivue, Inc and receives consulting fees. The other authors have nothing to disclose. The authors take full responsibility for the data and have the right to publish all data. This does not alter our adherence to PLOS ONE policies on sharing data and materials.

## Results

The HB9 had strong psychometric properties with a Cronbach α of 0.898 (95%CI: 0.882-0.913) and low floor and ceiling effects. The HB9 performed well across different sociodemographic groups. Lower HB9 scores were associated with greater resilience, better physical performance, and less physical frailty. Higher HB9 scores were associated with more comorbid medical conditions, more mood symptoms, lower resilience, and more functional impairment. A cut-off score of 4 on the HB9 provided a 15-fold ability to detect SCD in cognitively normal individuals, and a 14-fold ability to detect anosognosia in MCI.

## Conclusions

The use of the HB9 as an assessment of subjective cognitive complaints may help identify SCD for potential interventions and enrollment into clinical trials. The HB9 may also identify anosognosia which could lead to worse outcomes in MCI.

## Introduction

Subjective cognitive decline (SCD) represents the self-reported experience of declining cognitive functioning compared with the individual's perceived previous state [1]. SCD affects ~10% of adults ≥ 45y across different ethnoracial groups (5.0% Asian/Pacific Islanders, 9.3% non-Hispanic Whites, 10.1% African Americans, 11.4% Hispanics, 16.7% American Indians/Alaska Natives) [2]. SCD has been deemed a growing public health concern and a 'call to action' by the Center for Disease Control and Prevention encouraged screening for SCD during routine medical visits [3].

Risk factors for SCD include older age, female sex, anemia, thyroid diseases, lack of physical exercise, living alone, lack of health insurance, lower educational attainment, mood disorders, and sleep disorders [1,4,5]. Although standardized criteria exist, SCD may be measured differently including different timeframes (e.g., past year vs. 10 years) or comparison groups (e.g., individual's perceived baseline vs. peers). SCD has been linked to higher risk of mild cognitive impairment (MCI) and Alzheimer's disease (AD) [1]. Although not all people with SCD will develop objective cognitive change and subsequent dementia, many do. Individuals with SCD have been reported as having a higher presence of an APOE ε4 allele [6], increased risk of amyloid deposition [7], and a two-fold increase in risk of MCI [8,9] compared to individuals without SCD. Thus, identification of SCD could possibly improve early detection of MCI and promote early intervention or enrollment in clinical trials.

By definition, individuals with SCD have normal cognition [1], however individuals with MCI may also have subjective cognitive complaints [9,10]. Patients with SCD and MCI may have no differences in the degree of self-reported memory complaints [11]. Conversely, some patients with MCI experience anosognosia which is the lack of awareness or denial of a cognitive deficit [12,13]. A recent study found that while nearly half of patients with AD have anosognosia, it was much less common in MCI

affecting ~10% of patients [14,15]. While MCI with anosognosia tends to have poor overall cognitive performance, only small-to-moderate correlations between subjective complaints and neuropsychological tests or biomarkers have been described [16,17].

Several measures of SCD exist, each with strengths and limitations. Ideally, a SCD measure should evaluate a person's self-reported perception of cognitive decline and could be used as a screening tool to identify individuals who might be at higher risk for developing MCI or dementia [18]. Many of the available SCD instruments have not been sufficiently studied in diverse community-based samples, and those applied in community-based samples have not necessarily been validated against objective cognitive performance or biomarkers of neurodegenerative disease. There are few instruments designed to assess anosognosia in MCI or dementia. The Clinical Insight Rating Scale [19] and Abridged Anosognosia Questionnaire-Dementia [20] have been proposed in a recent review [21], but there remains no consensus how to assess either SCD or anosognosia. Further, as current instruments were developed to specifically characterize either SCD or anosognosia, there is no literature to suggest how an SCD instrument would perform in MCI with anosognosia, or how an instrument designed to quantify anosognosia in dementia would capture SCD. To address these unmet needs, we developed the Healthy Brain 9 (HB9) as a brief, self-report assessment of cognitive functioning and whether self-recognized changes interfere with daily functioning. We hypothesized that the HB9 would characterize SCD and identify anosognosia in MCI.

## Materials and methods

### Formative development of the HB9

The HB9 was developed following the findings of three studies from our group. The first used Alzheimer's Disease Neuroimaging Initiative (ADNI) data, optimized feature sets, and machine learning to identify items from the patient and study partner versions of the Everyday Cognition Scale (ECog) [22] and study partner reported Functional Activities Questionnaire (FAQ) [23] that assisted in the classification of early-stage AD [24]. The second used ADNI data cross-validated against a memory disorder clinic to identify items from the patient and study partner ECog, FAQ, and patient and study partner Quick Dementia Rating System (QDRS) [25,26] to screen for early-stage AD [27]. The third was a retrospective review of self-reported items from semi-structured interviews to generate a Clinical Dementia Rating (CDR) [28] in older adults enrolled in a study of brain aging [29]. These three analyses identified 9 themes of self-reported items that identified the earliest suggestions of cognitive change noticed by the participants. Questions were refined through several rounds of review to create the final 9-item scale (Fig 1) containing self-reports of cognitive abilities and everyday functioning using a reference period comparing current performance to respondents' abilities 5 years prior. Five years was chosen as the timeframe because short time frames (i.e., the "past year") were felt to potentially be more sensitive to any acute illnesses or life events, and longer time frames (i.e., "10 years") were felt to be too long of a window to be predictive of future objective cognitive impairment. The HB9 is scored with a 5-point Likert scale anchored from No Change to Marked Change (range of scores from 0–36) with higher scores reflecting greater subjective cognitive complaints. The HB9 takes on average 2 minutes to complete. The HB9 was then embedded into our ongoing longitudinal study of brain health for testing and validation.

### Study participants

Participants were 344 consecutive participants enrolled in the Healthy Brain Initiative (HBI) at the University of Miami Miller School of Medicine. The complete protocol has been published [30] but is briefly summarized here. Inclusion criteria include community-residing men and women aged 50y+ with a diagnosis of no cognitive impairment (NCI), SCD, or MCI who could (a) provide informed consent; (b) identify a study partner (c) undergo an MRI; and (d) speak, read, write, and understand English or Spanish. Exclusion criteria were unstable or significant illnesses that could affect participation in

**THE HEALTHY BRAIN 9 (HB9)**

The following descriptions characterize changes you may have noticed in your memory, thinking and everyday activities. The important thing to consider is comparing your abilities **now** with how they **used** to be.

Compared to **5 years ago,** please rate your current ability using the following scale
   **No Change:** I can do these things the same as I ever could
   **Very Mild Change:** I notice some differences, but they are inconsistent or do not cause any difficulties
   **Mild Change:** Differences are noticeable to me, but maybe not to others
   **Moderate Change:** Definite differences, and others may notice the change
   **Marked Change:** I now have trouble doing these things for myself and need assistance from others

Please Choose **ONE ANSWER** for each question

| Compared to **5 years ago,** please rate your current ability on the following: | No Change (0) | Very Mild Change (1) | Mild Change (2) | Moderate Change (3) | Marked Change (4) |
|---|---|---|---|---|---|
| Remembering recent events, conversations, phone messages, appointments, and/or taking my medications | | | | | |
| Making decisions, solving everyday problems, and being able to focus and concentrate at my expected level of performance | | | | | |
| Keeping accurate track of the correct date and day of the week | | | | | |
| Taking care of financial matters such as paying bills, managing money and bank accounts, and keeping track of business papers | | | | | |
| Carrying out my everyday activities (work, shopping, community activities, religious services, social groups) at my usual level of performance | | | | | |
| Having conversations, finding my words, communicating with others, reading, and writing at my usual level of performance | | | | | |
| Doing my household chores and tasks (such as cooking, cleaning, gardening, home repairs), and operating appliances to my usual standards | | | | | |
| Traveling independently outside my local neighborhood without having to use directions or getting mixed up | | | | | |
| Doing my hobby and recreational activities and keeping up with personal interests at my usual pace and level | | | | | |
| **TOTALS** | | | | | |

The Healthy Brain 9 (HB9) was created by James E. Galvin, MD, MPH and is copyrighted by the University of Miami Miller School of Medicine. All rights reserved.

**Fig 1. The Healthy Brain 9 (HB9).**

brain imaging or that may produce unreliable cognitive performance (e.g., metastatic cancer, poorly controlled diabetes). Recruitment was done through community outreach, educational programs, and word-of-mouth. Participants were remunerated for their time. Participants underwent comprehensive evaluations modelled on the Uniform Data Set (UDS v3.0) from the NIA Alzheimer's Disease Center program [31,32]. All components of the HBI evaluation were translated and

bilingual research staff conducted assessments in Spanish for Spanish-speaking participants who preferred to be tested in their native tongue. Participants completed the HB9 using an online survey (REDCap) prior to their in-person clinical visit. The HB9 was not used in the diagnostic process. At the end of the research assessment, participants met with the research clinician to review findings and recommendations for brain health and clinical follow-up, if required. The Healthy Brain Initiative, an ongoing longitudinal study, was first approved by the University of Miami's Institutional Review Board (Reference # 20200208) on 05/7/2020 with the most recent continuing review approved on 03/14/2025 and effective through 01/23/2026. All participants provided written informed consent. The first participant included in the current paper was enrolled on 03/24/2022 and the last participant included was enrolled on 02/04/2025.

## Clinical assessment

Sociodemographic data, primary language, medical history, medications, alcohol/tobacco/substance use history, and family history were collected. A detailed clinical and neurological examination including the Movement Disorder Society-Unified Parkinson's Disease Rating Scale, Part III Motor (MDS-UPDRS) [33] was completed. Every participant had an overnight home-based sleep study using WatchPAT One (ZOLL Itamar, Israel) to assess for obstructive sleep apnea. The Charlson Comorbidity Index [34] measured overall health and medical comorbidities. The Hospital Anxiety and Depression scale (HADS) [35] captured distinct ratings of depression and anxiety. The Resilience Index (RI) [36], a summed total of 6 protective factors (cognitive reserve, physical, cognitive and social activities, diet, mindfulness), was used to estimate cognitive resilience. The Vulnerability Index (VI) [37], a 12-item weighted measure of 8 modifiable (diabetes, heart disease, stroke, hypertension, hypercholesterolemia, obesity, frailty, depression) and 4 non-modifiable (age, sex, race/ethnicity, education) risk factors associated with the development of cognitive impairment, was used to assess future risk of dementia. The Modified Hachinski Scale [38] assessed the risk of vascular cognitive impairment. The mini-Physical Performance Evaluation (mPPT) [39] and Fried Frailty Phenotype [40] assessed physical functionality and frailty. Study partners completed a semi-structured interview with an experienced research clinician to derive the CDR and its sum of boxes (CDR-SB) [28]. The study partners also completed the QDRS-informant version, FAQ, and Neuropsychiatric Inventory Questionnaire (NPI-Q) [41].

## Neuropsychological assessment

Participants completed the QDRS – patient version for a subjective rating of cognitive status. The Montreal Cognitive Assessment (MoCA) [42] was administered for a global screen. The neuropsychological test battery included items from the UDS v3.0 [31,32] covering memory (Craft Story paragraph recall – verbatim and paraphrase), language (Animal Naming and Multilingual Naming Test), executive function (Trailmaking A and B), working memory (Numbers Forward and Backward), and visual (Benton Figure Copy and Recall). The UDS v3.0 battery was supplemented with the Hopkins Verbal Learning Task (episodic memory for word lists – immediate, delayed and recognition) [43], and the Number-Symbol Coding Test (executive function) [44]. Raw scores for each test were converted into z-scores based on the sample's mean and standard deviation to create a global z-score. Participants also completed Cognivue *Clarity*, a 10-minute computerized cognitive battery, providing a global score (range 0–100) [45]. The CDR, CDR-SB, and the Global Deterioration Scale (GDS) [46] were used for global staging. While both the CDR and GDS rate normal cognition, MCI, and dementia, only the GDS contains a category for subjective cognitive impairment (GDS 2).

## Blood based biomarkers

Blood-based markers were analyzed using two commercially available platforms. PrecivityAD2 (C2N, St Louis, MO) [47] uses mass spectrometry to provide measures of Aβ40, Aβ42, Aβ42/Aβ40, ApoE ε4 proteotype, ptau217, ptau217%ratio (ptau217/non-ptau217), and the Amyloid Probability Score 2 (APS2). Simoa™SR-X (Quanterix Corporation, Billerica, MA) [48] uses an immunohistochemical approach to provide measures of ptau181, neurofilament light chain (NFL), glial acidic fibrillary protein (GFAP).

## Magnetic resonance imaging

MRI scans were performed using a GE 3T 750W scanner and included high-resolution 3D sagittal MPRAGE, axial FLAIR, and T2* sequences. Quantitative morphometry was assessed with Combinostics™ cMRI (Finland), an FDA-cleared AI pipeline, to provide cortical, ventricular, and subcortical volumes and white matter hyperintensity burden [49]. Cortical atrophy scores (CAS), generated as z-score estimates, assessed global cortical atrophy derived from the global cortical atrophy four-point visual rating scale and adjusted for head size, age, and sex based on normative data from a sample of individuals aged 50–90. Higher CAS scores indicate greater atrophy [50].

## Determination of cognitive status

Clinical and cognitive data (excluding the HB9) were consolidated using a clinical consensus conference to assign individuals to the following diagnostic categories: No Cognitive Impairment (NCI), SCD, or MCI. NCI individuals had normal objective cognitive performance, offered no subjective complaints during the semi-structured CDR interview, had QDRS scores ≤1.5 or a score of 0 in the memory domain, and were rated CDR 0 and GDS 1. SCD individuals had normal objective cognitive performance, endorsed subjective complaints that their cognition was declining during the semi-structured CDR interview, had scores 2 or greater on the QDRS or a score >0 in the memory domain, and were rated CDR 0 and GDS 2. MCI individuals had evidence of objective cognitive impairment (scoring >1.5 standard deviations below age- and education norms in at least one cognitive domain), preserved everyday functioning as reported by the study partner, and were rated CDR 0.5 and GDS 3. Subjective cognitive complaints were not considered in the assignment of MCI. In this study, 62% of MCI participants reported that they were aware that their cognition was declining (i.e., MCI with awareness). The remaining 38% of MCI participants were classified as MCI with anosognosia. MCI etiologies were assigned using their phenotypic presentation, history, and current published clinical criteria to diagnose MCI due to AD [51], dementia with Lewy bodies [52], or vascular cognitive impairment [53]. Individuals with MCI due to depression were assigned based on objective cognitive impairment, normal physical and neurological exam, and endorsement of depression symptoms with scores >7 on the HADS. Individuals with MCI due to obstructive sleep apnea were assigned based on objective cognitive impairment, normal physical and neurological exam, and abnormal apnea-hypopnea index scores >5 on their overnight sleep study. MRI and blood-based biomarkers were not used in the clinical consensus diagnosis.

## Statistical analyses

Statistical analyses were conducted using IBM SPSS v29 (Armonk, NY). Descriptive statistics were used to summarize overall sample characteristics. Independent sample t-tests or one-way analysis of variance (ANOVA) with Tukey-Kramer post-hoc tests were used for continuous data, and Chi-square analyses were used for categorical data. Multiple comparisons were addressed using the Bonferroni correction. Strength of association compared HB9 scores with other rating scales and test performance. Linear regression modeling was then used to identify predictors of higher HB9 scores. Internal consistency was examined as the proportion of variability in the responses resulting from differences in the respondents, reported as the Cronbach alpha reliability coefficient. Coefficients greater than 0.7 are good measures of internal consistency. Data completeness was assessed by calculating the missing data rates for each HB9 item. Item variability, item frequency distributions, ranges, and standard deviations were calculated, and ceiling and floor effects were determined.

Receiver operator characteristic (ROC) curves and optimal cut-points (using closest top-left criteria and Youden index) were used to assess discrimination of the HB9 between groups. Results were reported as the area under the curve (AUC) with 95% confidence intervals (CI). Positive likelihood ratio, negative likelihood ratio, and diagnostic odds ratio were reported. Likelihood ratios range from 0 to infinity, with larger numbers providing more convincing evidence of disease and smaller numbers indicating that disease is less likely. Unlike positive and negative predictive values, likelihood ratios are

independent of disease prevalence. The diagnostic odds ratio represents the odds of positivity in individuals with disease relative to the odds of individuals without disease, providing a measure of test effectiveness.

## Hypothesis testing

We addressed the following 4 research questions:

1. What are the psychometric properties of the HB9 using the entire cohort?

2. Does the HB9 characterize SCD compared to NCI?

3. Does the HB9 detect anosognosia in MCI compared to MCI participants with awareness of their cognitive deficits?

4. Are there differences in subjective cognitive complaints between SCD and MCI participants?

## Results

### Sample characteristics

We evaluated 344 participants who completed the HB9 and had clinical consensus diagnoses. The sample had a mean age of 68.5±9.3y and were 70% female. Educational attainment included 12.3% with 12 years or less, 50.1% with 13–16 years, and 37.5% with more than 16 years of education. The ethnoracial make-up of the sample was 62% non-Hispanic White, 15% Black or African American, 18% Hispanic, and 6% Other ethnoracial groups. The sample included 146 NCI (42%), 93 SCD (27%) and 105 MCI (30%) participants. The mean MoCA scores were 25.5±2.9 (range 17–30), the mean CDR-SB was 0.5±0.9 (range 0–6), and the mean QDRS-patient version scores were 1.1±1.4 (range 0–9). The mean HB9 scores were 4.1±4.7 (range 0–28). Sample characteristics by diagnosis are shown in Table 1.

**Research question #1: Psychometric properties of the HB9.** Table 2 demonstrates the item distribution and inter-item correlations for the HB9 considering the entire cohort. There were low floor (19.0%) and ceiling (0%) effects. The standard deviations were similar for all 9 items, with the greatest variance for question 1 (Remembering recent events, conversations, phone messages, appointments and/or taking medications). The individual HB9 items were moderately correlated with each other suggesting that each question covered different perceptions of cognitive functioning. However, each item strongly correlated with the overall HB9 score. Participants were able to answer the HB9 without difficulty and their responses covered the range of choices. Cronbach alpha was excellent at 0.898 (95%CI: 0.882–0.913).

Strength of association of the HB9 with Gold Standard measures of clinical, cognitive, functional, behavioral, and physical ratings, as well as MRI and blood-based biomarkers is shown in Table 3. After correction for multiple comparisons, the HB9 showed small-to-moderate correlations with most clinical measures. HB9 scores had large correlations with the QDRS-patient version but only small correlations with the QDRS-informant version. There were no differences in HB9 scores by age, sex, race, Hispanic ethnicity, or APOE carrier status. There were educational effects on HB9 scores. Post college graduates (n = 133) had lower HB9 scores (3.5±4.0 vs. 5.7±5.4; F = 3.3, df 2, 349, p = 0.037) compared to individuals with 12 or less years of education (n = 43). Individuals with 13–16 years of education (n = 177) scores were not different (4.7±6.0) from the other two education groups. Two of the HB9 questions were most sensitive to an educational effect – Question 2: Making decisions/solving problems (F = 6.5, df 2, 350, p = 0.002) and Question 4: Taking care of financial matters (F = 3.3, df 2, 349, p = 0.037).

The HB9 showed small-to-moderate correlations with neuropsychological tests of global function, language, executive-attention, list learning, and visual reproduction suggesting that the HB9 was not measuring objective cognitive performance but rather the participants' perception of their performance compared to how they perceived they would have done in the past. The HB9 was not correlated with blood-based biomarkers except for a weak correlation with NFL. The HB9 was weakly correlated with imaging markers including smaller medial temporal lobe and hippocampal volumes and higher cortical atrophy scores.

**Table 1. Sample characteristics by diagnosis.**

| | NCI | SCD | MCI | Statistic[1] | p-value[2] | df[1] |
|---|---|---|---|---|---|---|
| *Clinical Variables* | | | | | | |
| Age, y | 67.9 (8.7) | 67.1 (8.9) | 70.5 (10.2) | 3.9 | 0.020[b] | 2,341 |
| Education, y | 16.1 (2.9) | 15.9 (2.9) | 15.1 (3.4) | 3.4 | 0.034[a] | 2,341 |
| Sex, % Female | 74.5 | 72.0 | 61.9 | 4.8 | 0.089 | 2 |
| Race, % NHW | 68.3 | 60.9 | 53.8 | 10.9 | 0.090 | 6 |
| Memory complaints, % | 0.0 | 100.0 | 61.9 | 202.4 | <0.001[a,b,c] | 2 |
| QDRS-patient | 0.4 (0.8) | 1.6 (1.3) | 1.5 (1.8) | 29.8 | <0.001[a,c] | 2,338 |
| QDRS-informant | 0.4 (0.7) | 1.3 (1.7) | 1.1 (1.6) | 15.2 | <0.001[a,c] | 2,333 |
| CDR-SB | 0.04 (0.1) | 0.09 (0.3) | 0.8 (0.4) | 258.1 | <0.001[a,b] | 2,333 |
| FAQ | 0.3 (1.0) | 1.0 (2.3) | 0.8 (1.8) | 5.7 | 0.004[c] | 2,336 |
| NPI-Q | 1.0 (1.7) | 1.8 (2.7) | 1.9 (2.9) | 4.3 | 0.014[a,c] | 2,319 |
| HADS-Anxiety | 4.1 (2.8) | 5.5 (3.5) | 4.9 (3.4) | 5.9 | 0.003[c] | 2,333 |
| HADS-Depression | 2.8 (2.1) | 4.9 (3.6) | 4.9 (3.8) | 19.0 | <0.001[a,c] | 2,333 |
| Charlson Comorbidity | 0.5 (0.8) | 0.6 (1.0) | 1.4 (1.5) | 17.6 | <0.001[a,b] | 2,339 |
| Hachinski | 0.5 (0.6) | 0.5 (0.7) | 0.7 (0.8) | 7.8 | <0.001[a] | 2,336 |
| mPPT | 12.7 (2.5) | 12.2 (2.2) | 11.0 (2.6) | 8.9 | <0.001[a,b] | 2,335 |
| Fried Frailty Phenotype | 0.7 (0.9) | 0.9 (1.0) | 1.3 (1.2) | 15.1 | <0.001[a,b] | 2,341 |
| MDS-UPDRS | 2.6 (4.4) | 2.3 (3.8) | 7.0 (9.8) | 13.0 | <0.001[a,b] | 2,339 |
| VI | 6.8 (2.4) | 7.1 (2.3) | 8.2 (2.7_ | 10.5 | <0.001[a,b] | 2,334 |
| RI | 181.2 (31.9) | 174.1 (30.7) | 164.8 (27.9) | 8.8 | <0.001[a] | 2.335 |
| MoCA | 26.8 (2.3) | 26.2 (2.3) | 22.9 (2.8) | 80.8 | <0.001[a,b] | 2,341 |
| Cognivue | 78.5 (9.9) | 78.4 (8.7) | 62.7 (15.9) | 58.9 | <0.001[a,b] | 2,335 |
| Global z-score | 0.25 (0.4) | 0.25 (0.4) | −0.58 (0.5) | 131.7 | <0.001[a,b] | 2,326 |
| HB9 | 1.9 (2.1) | 5.9 (4.7) | 5.5 (5.9) | 33.6 | <0.001[a,c] | 2,341 |
| *Plasma and Imaging Biomarkers* | | | | | | |
| pTau217% | 1.5 (1.4) | 2.0 (2.7) | 1.7 (1.8) | 2.0 | 0.137 | 2,303 |
| pTau181 | 22.9 (11.7) | 23.7 (10.4) | 25.2 (10.1) | 1.2 | 0.368 | 2,323 |
| NFL | 12.4 (6.7) | 11.9 (5.8) | 14.4 (8.9) | 3.1 | 0.044 | 2,313 |
| GFAP | 186.2 (97.2) | 190.4 (95.2) | 227.5 (146.4) | 4.0 | 0.018[a] | 2,313 |
| WMH volume | 3.9 (6.9) | 4.7 (7.3) | 6.9 (9.1) | 3.6 | 0.030[a] | 2,250 |
| Hippocampal volume | 7.5 (0.8) | 7.5 (0.9) | 6.9 (0.9) | 11.8 | <0.001[a,b] | 2,253 |
| Ventricular volume | 30.9 (15.2) | 36.2 (17.9) | 42.7 (23.0) | 9.4 | <0.001[a] | 2,255 |
| CAS | 1.3 (0.9) | 1.5 (0.9) | 1.9 (0.9) | 9.5 | <0.001[a,b] | 2,253 |

Mean (SD) or %.

Key: NCI = No Cognitive Impairment, SCD = Subjective Cognitive Decline, MCI = Mild Cognitive Impairment, NHW = Non-Hispanic White, QDRS = Quick Dementia Rating System, CDR-SB = Clinical Dementia Rating Sum of Boxes, FAQ = Functional Activities Questionnaire, NPI-Q = Neuropsychiatric Inventory-Questionnaire, HADS = Hospital Anxiety and Depression Scale, mPPT = mini Physical Performance Scale, MDS-UPDRS = Movement Disorder Society-Unified Parkinson's Disease Rating Scale-Part III Motor, VI = Vulnerability Index, RI = Resilience Index, MoCA = Montreal Cognitive Assessment, NFL = Neurofilament Light Chain, GFAP = Glial Fibrillary Acidic Protein, CAS = Cortical Atrophy Score.

[1]F-statistic and degrees of freedom from one-way ANOVA (Continuous) or $c^2$ statistic and degrees of freedom from Chi-Square (categorical).

[2]p-value corrected for multiple comparisons using Bonferroni methods (corrected p-value = 0.0022 for clinical variables and p = 0.00625 for biomarkers).

Tukey post-hoc comparisons.

[a]NCI different from MCI.

[b]SCD different from MCI.

[c]NCI different from SCD.

**Table 2. HB9 item distributions, inter-item, and item-total correlations.**

| HB9 Question | Item Distribution and Missing Rates | | | | | | | | | Item-Total Pearson r |
| --- | --- | --- | --- | --- | --- | --- | --- | --- | --- | --- |
| | Item | | Response counts% | | | | | | | |
| | Mean | SD | No Change | Very Mild Change | Mild Change | Moderate Change | Marked Change | Missing | | |
| Remembering recent events, conversations, phone messages, appointments, and/or taking my medications (Q1) | 0.978 | 0.969 | 36.3 | 39.7 | 15.6 | 6.7 | 1.7 | 0.0 | | 0.734 |
| Making decisions, solving everyday problems, and being able to focus and concentrate at my expected level of performance (Q2) | 0.559 | 0.860 | 62.6 | 24.6 | 7.8 | 4.5 | 0.6 | 0.0 | | 0.807 |
| Keeping accurate track of the correct date and day of the week (Q3) | 0.391 | 0.672 | 69.8 | 22.9 | 5.6 | 1.7 | 0.0 | 0.0 | | 0.653 |
| Taking care of financial matters such as paying bills, managing money and bank accounts, and keeping track of business papers (Q4) | 0.310 | 0.691 | 78.5 | 15.1 | 3.9 | 2.0 | 0.6 | 0.0 | | 0.739 |
| Carrying out my everyday activities (work, shopping, community activities, religious services, social groups) at my usual level of performance (Q5) | 0.260 | 0.637 | 81.8 | 12.8 | 3.1 | 2.0 | 0.3 | 0.0 | | 0.797 |
| Having conversations, finding my words, communicating with others, reading, and writing at my usual level of performance (Q6) | 0.735 | 0.893 | 48.6 | 35.5 | 11.5 | 2.8 | 1.7 | 0.0 | | 0.717 |
| Doing my household chores and tasks (such as cooking, cleaning, gardening, home repairs), and operating appliances to my usual standards (Q7) | 0.391 | 0.794 | 74.3 | 17.3 | 4.5 | 2.8 | 1.1 | 0.0 | | 0.821 |
| Traveling independently outside my local neighborhood without having to use directions or getting mixed up (Q8) | 0.325 | 0.769 | 79.1 | 14.0 | 3.4 | 1.7 | 1.7 | 0.3 | | 0.692 |
| Doing my hobby and recreational activities and keeping up with personal interests at my usual pace and level (Q9) | 0.441 | 0.813 | 69.8 | 21.2 | 5.6 | 1.7 | 1.7 | 0.0 | | 0.771 |
| **Interitem Correlation Matrix** | **Q1** | **Q2** | **Q3** | **Q4** | **Q5** | **Q6** | **Q7** | **Q8** | **Q9** | |
| Remembering recent events, conversations, phone messages, appointments, and/or taking my medications (Q1) | 1 | | | | | | | | | |
| Making decisions, solving everyday problems, and being able to focus and concentrate at my expected level of performance (Q2) | 0.616 | 1 | | | | | | | | |
| Keeping accurate track of the correct date and day of the week (Q3) | 0.431 | 0.454 | 1 | | | | | | | |
| Taking care of financial matters such as paying bills, managing money and bank accounts, and keeping track of business papers (Q4) | 0.478 | 0.607 | 0.498 | 1 | | | | | | |
| Carrying out my everyday activities (work, shopping, community activities, religious services, social groups) at my usual level of performance (Q5) | 0.458 | 0.603 | 0.475 | 0.637 | 1 | | | | | |
| Having conversations, finding my words, communicating with others, reading, and writing at my usual level of performance (Q6) | 0.469 | 0.509 | 0.413 | 0.475 | 0.556 | 1 | | | | |
| Doing my household chores and tasks (such as cooking, cleaning, gardening, home repairs), and operating appliances to my usual standards (Q7) | 0.510 | 0.593 | 0.473 | 0.523 | 0.634 | 0.500 | 1 | | | |

*(Continued)*

**Table 2.** (Continued)

| | Item Distribution and Missing Rates | | | | | | | | Item-Total Pearson r |
|---|---|---|---|---|---|---|---|---|---|
| | Item | | Response counts% | | | | | | |
| HB9 Question | Mean | SD | No Change | Very Mild Change | Mild Change | Moderate Change | Marked Change | Missing | |
| Traveling independently outside my local neighborhood without having to use directions or getting mixed up (Q8) | 0.409 | 0.467 | 0.356 | 0.390 | 0.509 | 0.372 | 0.591 | 1 | |
| Doing my hobby and recreational activities and keeping up with personal interests at my usual pace and level (Q9) | 0.432 | 0.539 | 0.400 | 0.458 | 0.594 | 0.472 | 0.716 | 0.586 | 1 |

Key: HB9 = Healthy Brain 9.

**Table 3. Concurrent validity of HB9 with clinical, cognitive and biomarker variables.**

| Clinical Variable | R[1] | p-value[2] | Cognitive Variable | R[1] | p-value[2] |
|---|---|---|---|---|---|
| Age | 0.018 | 0.738 | MoCA | −0.189 | <0.001 |
| Education | 0.146 | 0.006 | Numbers Forward | −0.026 | 0.652 |
| QDRS – patient version | 0.716 | <0.001 | Numbers Backward | −0.081 | 0.163 |
| QDRS – informant version | 0.278 | <0.001 | Trailmaking A (time) | 0.173 | 0.003 |
| CDR-SB | 0.278 | <0.001 | Trailmaking A (mistakes) | 0.306 | <0.001 |
| Functional Activities Questionnaire | 0.301 | <0.001 | Trailmaking B (time) | 0.176 | 0.002 |
| Neuropsychiatric Inventory | 0.195 | <0.001 | Trailmaking B (mistakes) | 0.138 | 0.017 |
| HADS – Anxiety | 0.322 | <0.001 | Animal Naming | −0.161 | 0.005 |
| HADS – Depression | 0.411 | <0.001 | MINT | −0.225 | <0.001 |
| Charlson Comorbidity Index | 0.259 | <0.001 | Number Symbol Coding Test | −0.211 | <0.001 |
| Hachinski | 0.136 | 0.017 | HVLT – Immediate Recall | −0.159 | 0.006 |
| mPPT | −0.214 | <0.001 | HVLT – Delayed Recall | −0.162 | 0.005 |
| Fried Frailty Phenotype | 0.235 | <0.001 | HVLT – Recognition | −0.138 | 0.016 |
| MDS-UPDRS | 0.083 | 0.145 | Craft Story – Immediate Recall (Verbatim) | −0.027 | 0.639 |
| Vulnerability Index | 0.187 | 0.001 | Craft Story – Immediate Recall (Paraphrase) | −0.014 | 0.809 |
| Resilience Index | −0.391 | <0.001 | Craft Story – Delayed Recall (Verbatim) | −0.089 | 0.123 |
| **Blood Based and MRI Biomarker** | **R[1]** | **p-value[2]** | Craft Story – Delayed Recall (Paraphrase) | −0.077 | 0.185 |
| pTau217%ratio | 0.071 | 0.244 | Benton Figure Copy | −0.174 | 0.002 |
| pTau181 | 0.033 | 0.565 | Benton Figure Recall | −0.114 | 0.048 |
| NFL | 0.156 | 0.008 | Global z-score | −0.201 | <0.001 |
| GFAP | 0.059 | 0.318 | Cognivue *Clarity* | −0.233 | <0.001 |
| White matter hyperintensity volume, ml | 0.095 | 0.158 | | | |
| Cortical Atrophy Score | 0.228 | <0.001 | | | |
| Hippocampus volume, ml | −0.189 | 0.004 | | | |
| Total Ventricular volume, ml | 0.167 | 0.012 | | | |

[1]Pearson R.

[2]Corrected p-values = 0.0024 for clinical/cognitive variables and p = 0.01 for blood-based and MRI biomarkers.

Key: HB9 = Healthy Brain 9; QDRS = Quick Dementia Rating System; CDR-SB = Clinical Dementia Rating Sum of Boxes; HADS = Hospital Anxiety and Depression Scale; mPPT = mini–Physical Performance Test; MDS-UPDRS = Movement Disorder Society-Unified Parkinson's Disease Rating Scale-Part III Motor; NFL = Neurofilament Light Chain; GFAP = Glial Acidic Fibrillary Protein; MoCA = Montreal Cognitive Assessment; MINT = Multilingual Naming Test; HVLT = Hopkins Verbal Learning Test.

We further explored the biomarkers with subgroup analyses. For the NCI group, higher HB9 scores were correlated with higher Aβ42 levels (r = 0.239, p = 0.009) and higher NFL (r = 0.263, p = 0.003). For the SCD group, higher HB9 scores were marginally correlated with smaller medial temporal lobe (r = −0.269, p = 0.052) and hippocampal (r = −0.251, p = 0.070) volumes. For MCI with awareness, there were no significant correlations with any biomarker, while HB9 scores in MCI with anosognosia were strongly correlated with APS2 scores (r = 0.609, p < 0.001) and ptau217% ratios (r = 0.788, p < 0.001) suggesting the presence of more AD-related neuropathologic changes.

**Research question #2: HB9 performance detecting SCD.** We compared the performance of the HB9 between GDS 1 and GDS 2 individuals to assess its ability to characterize SCD (Table 4). In addition to total HB9 scores being different between groups, answers to each of the 9 individual questions were also significantly different with the largest differences in Question 1; Remembering recent events (t = −10.2, df = 237, p < 0.001), Question 2: Making decisions (t = −9.2, df = 237, p < 0.001) and Question 6: Having conversations (t = −7.6, df = 237, p < 0.001). Differences in HB9 reflect significant differences between GDS 1 and GDS 2 individuals in Resilience Index (t = −2.8, df = 232, p = 0.005), both QDRS-patient (t = −9.3, df = 235, p < 0.001) and QDRS-informant (t = −5.4, df = 231, p < 0.001), FAQ (t = −4.1, df = 232, p < 0.001), HADS-Depression (t = −5.9, df = 234, p < 0.001), and HADS-Anxiety (t = −3.5, df = 234, p < 0.001) scores. The only biomarker difference was a greater ptau217% (1.5 ± 1.4 vs. 2.1 ± 2.7; t = −2.2, df = 217, p = 0.026) in SCD. Linear regression models demonstrated that predictors of higher HB9 scores in cognitive normal controls include higher scores on HADS-Depression (β = 0.412, 95%CI:0.365–0.723, p < 0.001), Vulnerability Index (β = 0.152, 95%CI:0.041–0.512, p = 0.022), and FAQ (β = 0.153, 95%CI: 0.048–0.669, p = 0.024).

**Research question #3: HB9 performance identifying MCI with anosognosia.** We then compared the performance of the HB9 between MCI (GDS 3) participants with awareness of cognitive deficits (n = 65) compared with MCI with anosognosia (n = 40) as determined by consensus (Table 5). After correction for multiple comparisons, the total HB9 scores and the 5 of 9 individual items were significantly different. Items that were not different were Question 5 (Carrying

**Table 4. HB9 performance comparing NCI to SCD participants.**

| | NCI | SCD | Statistic[1] | p-value[2] | df[1] |
|---|---|---|---|---|---|
| HB9 Total Score | 1.7 (1.8) | 6.1 (4.7) | −10.2 | <0.001 | 236 |
| Remembering recent events, conversations, phone messages, appointments, and/or taking my medications (Q1) | 0.5 (0.7) | 1.4 (0.90 | −9.2 | <0.001 | 237 |
| Making decisions, solving everyday problems, and being able to focus and concentrate at my expected level of performance (Q2) | 0.2 (0.4) | 0.7 (0.9) | −6.7 | <0.001 | 237 |
| Keeping accurate track of the correct date and day of the week (Q3) | 0.1 (0.3) | 0.5 (0.7) | −5.9 | <0.001 | 237 |
| Taking care of financial matters, such as paying bills, managing money and bank accounts, and keeping track of business papers (Q4) | 0.05 (0.3) | 0.4 (0.7) | −5.6 | <0.001 | 237 |
| Carrying out my everyday activities (work, shopping, community activities, religious services, social groups) at my usual level of performance (Q5) | 0.05 (0.2) | 0.3 (0.6) | −4.7 | <0.001 | 237 |
| Having conversations, finding my words, communicating with others, reading, and writing at my usual level of performance (Q6) | 0.4 (0.5) | 1.1 (0.8) | −7.6 | <0.001 | 237 |
| Doing my household chores and tasks (such as cooking, cleaning, gardening, home repairs), and operating appliances to my usual standards (Q7) | 0.1 (0.4) | 0.5 (0.8) | −5.0 | <0.001 | 237 |
| Traveling independently outside my local neighborhood without having to use directions or getting mixed up (Q8) | 0.1 (0.3) | 0.5 (0.8) | −4.9 | <0.001 | 236 |
| Doing my hobby and recreational activities and keeping up with personal interests at my usual pace and level (Q9) | 0.2 (0.4) | 0.6 (0.8) | −4.9 | <0.001 | 237 |

Mean (SD).

[1]t-statistic and degrees of freedom from independent samples t-test.

[2]p-value corrected for multiple comparisons using Bonferroni methods (corrected p-value = 0.0055).

**Table 5. HB9 performance comparing MCI with and without anosognosia.**

| | MCI with Awareness | MCI with Anosognosia | Statistic[1] | p-value[2] | df[1] |
|---|---|---|---|---|---|
| HB9 Total | 7.3 (5.8) | 2.5 (4.8) | −4.4 | <0.001 | 103 |
| Remembering recent events, conversations, phone messages, appointments, and/or taking my medications (Q1) | 1.3 (1.0) | 0.5 (0.8) | −4.1 | <0.001 | 103 |
| Making decisions, solving everyday problems, and being able to focus and concentrate at my expected level of performance (Q2) | 1.1 (0.9) | 0.4 (0.8) | −3.6 | <0.001 | 103 |
| Keeping accurate track of the correct date and day of the week (Q3) | 0.7 (0.8) | 0.2 (0.5) | −3.8 | <0.001 | 103 |
| Taking care of financial matters, such as paying bills, managing money and bank accounts, and keeping track of business papers (Q4) | 0.5 (0.7) | 0.1 (0.4) | −3.5 | <0.001 | 103 |
| Carrying out my everyday activities (work, shopping, community activities, religious services, social groups) at my usual level of performance (Q5) | 0.5 (0.8) | 0.1 (0.6) | −2.6 | 0.011 | 103 |
| Having conversations, finding my words, communicating with others, reading, and writing at my usual level of performance (Q6) | 1.1 (1.0) | 0.4 (0.7) | −3.7 | <0.001 | 103 |
| Doing my household chores and tasks (such as cooking, cleaning, gardening, home repairs), and operating appliances to my usual standards (Q7) | 0.7 (0.9) | 0.2 (0.7) | −2.7 | 0.007 | 103 |
| Traveling independently outside my local neighborhood without having to use directions or getting mixed up (Q8) | 0.5 (0.9) | 0.2 (0.8) | −1.8 | 0.071 | 103 |
| Doing my hobby and recreational activities and keeping up with personal interests at my usual pace and level (Q9) | 0.8 91.0) | 0.3 (0.7) | −2.6 | 0.011 | 103 |

Mean (SD).

[1]t-statistic and degrees of freedom from independent samples t-test.

[2]p-value corrected for multiple comparisons using Bonferroni methods (corrected p-value=0.0055).

out everyday activities), Question 7 (Household chores), Question 8 (Traveling independently), and Question 9 (Hobbies). These finding suggest that MCI participants with awareness recognize cognitive but not functional decline, while MCI with anosognosia recognize neither. Significant differences between MCI with awareness vs MCI with anosognosia in clinical features were HADS-Depression (t=−3.2, df=98, p=0.001), and HADS-Anxiety (t=−2.6, df=98, p=0.009) scores, while differences in biomarkers were limited to GFAP (t=−2.2, df=90, p=0.033). Linear regression models demonstrated that predictors of higher HB9 scores in MCI with awareness lower Resilience Index scores (β=−0.374, 95%CI:-0.035–0.123, p<0.001) and higher Charlson Comorbidity Index scores (β=0.260, 95%CI: 0.199–1.735, p=0.014). Proportions of awareness of deficit and anosognosia were not different between AD and non-AD etiologies of MCI (χ²=5.2, df=2, p=0.074).

**Research question #4: Comparison of SCD and MCI with awareness of cognitive deficit.** No differences were found in total HB9 scores between SCD and MCI with awareness (t=−1.5, df=151, p=0.145). Upon examination of individual items, Question 2 (Making decisions) was different between SCD and MCI with awareness (0.7±0.9 vs. 1.1±1.0; t=−2.7, df=152, p=0.003).

## Discriminability of the HB9

We tested the discriminative properties of the HB9 to (a) capture subjective cognitive complaints across the entire sample, (b) detect SCD in cognitively normal participants, and (c) detect anosognosia in individuals with MCI (Table 6). Area under curve in each scenario demonstrated good discrimination of the HB9 to identify subjective impairment with a cut-off score of 3.5. Individuals with scores of 4 or greater endorse subjective impairment, while individuals with scores of 3 or less deny subjective impairment. While the sensitivity of the HB9 was less than desirable, there was good specificity for discriminating NCI from SCD and for detecting anosognosia in MCI. The sensitivity of any instrument defines the ability to

 

**Table 6. Receiver operator characteristics curve features of HB9.**

| | AUC (95% CI) | p-value | Sensitivity | Specificity | LR+ | LR- | DOR |
|---|---|---|---|---|---|---|---|
| Whole Sample | | | | | | | |
| | 0.833 (0.790-0.876) | <0.001 | 67.7 | 84.2 | 4.27 | 0.38 | 11.2 |
| Detecting SCD (endorsing 4 or more items in cognitively normal controls) | | | | | | | |
| | 0.833 (0.781-0.884) | <0.001 | 56.3 | 92.3 | 7.31 | 0.47 | 15.5 |
| Detecting Anosognosia (endorsing 3 or less items in mild cognitive impairment) | | | | | | | |
| | 0.816 (0.727-0.905) | <0.001 | 74.5 | 82.5 | 4.31 | 0.30 | 14.4 |

Key: AUC = Area Under the Curve; CI = Confidence Interval; LR+=Likelihood Ratio of a Positive Test; LR-Likelihood Ratio of a Negative Test; DOR = Diagnostic Odds Ratio.

capture true positives while the specificity defines the ability to identify true negatives. For the HB9, specificity is the more useful property since for SCD since we are trying to eliminate the NCI individuals. Similarly, for the study of anosognosia, we are trying to eliminate MCI with awareness. A better way of capturing this clinically is to use diagnostic odds ratios based on the positive and negative likelihood ratios. The diagnostic odds ratio to capture any subjective complaints was 11.2. The diagnostic odds ratio to detect SCD was 15.5, while the diagnostic odds ratio to detect Anosognosia in MCI was 14.4.

## Comparison of HB9 scores with research clinician gold standard

Finally, we compared HB9 scores with the clinician's final independent determination of whether subjective cognitive complaints were endorsed (Fig 2). There were significant differences between groups (F = 35.1, df = 3, 339, p < 0.001). Participants with NCI (1.7 ± 1.8) had the lowest HB9 scores while MCI with awareness (7.3 ± 5.8) had the highest HB9 scores.

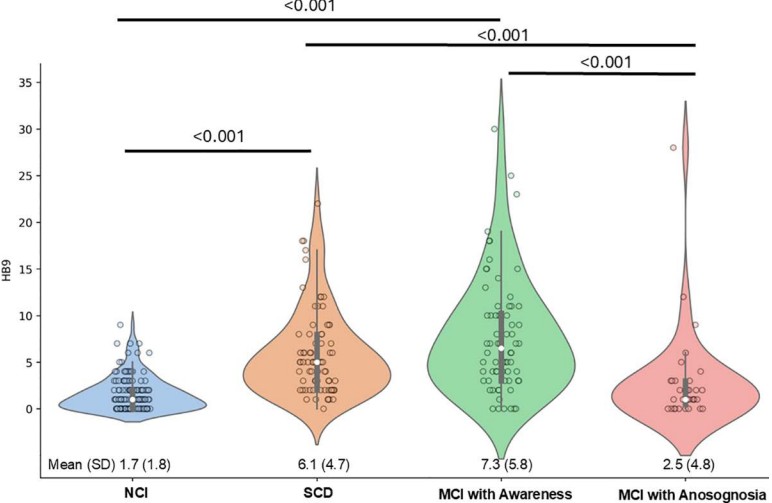

**Fig 2. Comparison of HB9 Scores with Research Clinician Global Assessment.** Self-reported subjective cognitive complaints on the HB9 were compared across clinical diagnostic groups (NCI, SCD, MCI with Awareness, and MCI with Anosognosia). Participants with NCI (1.7 ± 1.8) had the lowest HB9 scores while MCI with awareness (7.3 ± 5.8) had the highest HB9 scores. There was no difference in HB9 scores between NCI participants and MCI participants with anosognosia (2.5 ± 4.8), and no difference between SCD (6.1 ± 4.7) and MCI with awareness.

There was no difference in HB9 scores between NCI participants and MCI participants with anosognosia (2.5±4.8), and no difference in HB9 scores between SCD (6.1±4.7) and MCI with awareness.

## Discussion

These data support that the HB9 could greatly facilitate identification of individuals with subjective cognitive complaints. We demonstrate two potential uses for the HB9. In cognitively normal individuals, the HB9 had a 15-fold ability to differentiate individuals with SCD, while in cognitively impaired individuals, the HB9 had a 14-fold ability to detect the presence of anosognosia. The HB9 does not measure objective performance suggesting that the HB9 captures aspects of intraindividual change noted by the participant, rather than inter-individual change noted by neuropsychological testing. Based on the findings presented, we believe that the HB9 measures the construct of cognition quite differently from traditional objective cognitive measurements because it assesses the participants' point of view of whether they perceive that their memory, thinking, and everyday activities have changed and if they have changed, how significant that change is impacting their lives. This is fundamentally different from objective performance on a neuropsychological test of a specific domain such as memory, language, or executive function compared to an age- and education-norm. This may explain why the HB9 can distinguish NCI from SCD while the neuropsychological tests cannot. The same may be true for the HB9 ability to discriminate MCI with awareness from MCI with anosognosia when neuropsychological tests cannot. Predictors of higher HB9 scores included lower resilience, greater number of comorbid medical conditions, more depressive symptoms, and more functional impairment. Individuals with SCD and MCI with awareness share similar complaints while MCI with and without anosognosia failed to recognize functional decline.

We saw little association between disease-specific biomarkers (Aβ, ptau, white matter hyperintensities) and HB9 scores. This is different from reports that used BIOFINDER [54] or ADNI [55] participants, possibly due to the higher representation of AD cases and less sociodemographic diversity in those samples. We did find weak associations with non-specific neurodegeneration markers such as plasma NFL, hippocampal volume, and cortical atrophy scores. This observation is consistent with reports of greater hippocampal volume loss or cortical thinning associated with SCD [56,57]. This is also supported by subgroup analyses where HB9 scores in the NCI group suggested healthier brains and HB9 scores in SCD and MCI with anosognosia were associated with neurodegenerative changes. Overall, our findings support the use of the HB9 to capture subjective complaints about cognition and everyday functioning regardless of cognitive status or diagnosis. However, as some individuals with cognitive impairment have anosognosia [58], the HB9 may also serve as a measure of insight and awareness of deficits in individuals with objective cognitive impairment.

There may be differential patterns of subjective cognitive complaints between AD and non-AD etiologies, between MCI and dementia, and between cognitively normal controls. Self-reported complaints were most commonly associated with memory, followed by language, executive function, and visuospatial domains [59,60]. In a meta-analysis, subjective cognitive complaints were (a) greater in controls than amnestic MCI and AD, (b) not significantly different between amnestic and non-amnestic MCI, and (c) least present in AD compared with both amnestic and non-amnestic MCI [61].

There are several instruments currently in use to characterize SCD. The Subjective Cognitive Decline Questionnaire is a 24-item tool that measures self-perceived cognitive decline in memory, language, and executive functions [62]. The McCusker Subjective Cognitive Impairment Inventory is a 46-item questionnaire assessing 6 cognitive domains [63]. The Cognitive Change Index is a 20-item tool [64], and the ECog is a 39-item tool, each assessing multiple cognitive domains. The Cognitive Function Instrument is a 14-item assessment of cognitive status referencing performance to one year prior [65]. Each of these instruments captures self-reported cognitive symptoms that are related to activities of daily living. Many scales do not consider other manifestations of SCD, lack diversity of research participants, fail to capture psychological or physical frailty, and may not be useful in capturing anosognosia, perhaps limiting their practical applications outside of AD research.

There are fewer instruments that permit assessment of anosognosia [19–21]. A recent study calculated an Awareness of Cognitive Decline measure which used the difference between participant scores and study partner scores on the ECog with a differential of 3 signifying a significant subjective complaint [59]. The authors found that MCI individuals had good agreement on cognitive symptoms with their study partners, while the AD participants had poor agreement suggesting anosognosia. Disadvantages of this approach include the requirement of study partners limiting practicality in the clinical setting, and the lack of thresholds for what defines SCD vs no complaints vs anosognosia.

There is little literature to suggest how instruments designed for SCD would work in MCI with and without anosognosia and even less information about how an anosognosia instrument could be used in cognitively normal controls. While the E-Cog has been examined in both SCD and in the calculation of the Awareness of Cognitive Decline measurement, this required information from both the patient and a knowledgeable informant which limits its utility in the clinical setting where study partners rarely accompany patients. The HB9 is brief (~2 minutes) and can capture the frequency and severity of subjective symptoms in both cognitively normal individuals and those with cognitive impairment across different etiologies. When combined with an objective cognitive assessment (e.g., MoCA, Cognivue, or neuropsychological test battery), it may be possible to discriminate NCI from SCD, and between MCI with anosognosia from MCI without anosognosia.

## Limitations

This study focused on cross-sectional analyses to describe the properties of the HB9, but longitudinal follow-up is ongoing and will be needed to determine whether the HB9 is useful for predicting future cognitive decline. Although the sample was diverse in terms of age, sex, race and ethnicity, the cohort was skewed towards higher educational attainment. We did find an influence of education on HB9 with less complaints in those with more education, in particular in relation to two executive function questions regarding problem solving and finances. While no participants met the threshold for major depressive disorder, there was an association between HB9 scores and mood complaints. Further research is needed to fully explore this relationship. Anxiety and depression can sometimes manifest as subjective cognitive complaints, so it is important to consider mental health factors when interpreting results. There is the potential for recall bias using the 5-year retrospective self-report period. This is inherent in any instrument requiring past reporting common to nearly all patient-reported outcomes. Future studies could include methods to mitigate recall bias such as ecological momentary assessment. The HB9 should not be used as a diagnostic tool. Subjective cognitive decline assessment alone cannot diagnose MCI or dementia; further evaluation with comprehensive neuropsychological testing and informant-based assessments are needed.

Strengths of the HB9 include the ability to use a single instrument to detect SCD in cognitively normal controls and anosognosia in cognitively impaired individuals. The measurement properties of the HB9 align with many components of the consensus-based standards for the selection of health measurement instruments (COSMIN) taxonomy [66] with 6 of 9 properties established (internal consistency, face validity, criterion validity, structural validity, cross-cultural validity, and hypothesis testing). Future research in the HB9 should focus on establishing intra-rater reliability and measurement error (testing the same rater with repeated administration) and responsiveness (measuring change in outcomes over time). Longitudinal studies will be needed to draw conclusions about causality and the progression of SCD or anosognosia over time.

## Conclusions

While some studies have failed to show a relationship between SCD and future dementia [1,67,68], there is converging evidence that SCD is a possible risk factor for future dementia with higher subjective complaints indicating the presence of cognitive changes at sub-clinical levels. Accurate and valid measurement of SCD is critical to identifying individuals that may have other preclinical markers for future MCI and dementia. The primary method for SCD detection is through questionnaires or interviews where individuals describe their perceived changes in cognitive abilities, focusing on areas

like memory, attention, language, and executive function. The HB9 captures both subjective cognitive complaints and how much this complaint may be interfering with everyday functioning. Use of the HB9 as an assessment of SCD may help identify individuals in the early stages of a neurodegenerative disease who could benefit from further monitoring, potential interventions, and enrollment into clinical trials. As the HB9 also detects MCI individuals with anosognosia, this has great value in the clinical setting to help better manage patients who lack insight and may be more likely to have lower adherence to medication use and clinician recommendations for care.

## Acknowledgments

The authors would like to thank the dedicated research participants and their study partners, faculty, staff, postdoctoral fellows, trainees of the Comprehensive Center for Brain Health at the University of Miami Miller School of Medicine.

## Author contributions

**Conceptualization:** James E. Galvin.

**Data curation:** James E. Galvin, Conor B. Galvin.

**Formal analysis:** James E. Galvin, Michael J. Kleiman.

**Funding acquisition:** James E. Galvin.

**Investigation:** Katherine C. Almonte, Andrea Buehler, Yolene M. Caicedo, Willman Jimenez, Mahesh S. Joshi, Nicole Mendez, Mary Lou A. Riccio, Marcia I. Walker.

**Methodology:** James E. Galvin, Michael J. Kleiman.

**Project administration:** James E. Galvin.

**Resources:** James E. Galvin.

**Supervision:** James E. Galvin.

**Writing – original draft:** James E. Galvin.

**Writing – review & editing:** James E. Galvin, Katherine C. Almonte, Andrea Buehler, Yolene M. Caicedo, Conor B. Galvin, Willman Jimenez, Mahesh S. Joshi, Nicole Mendez, Mary Lou A. Riccio, Marcia I. Walker, Michael J. Kleiman.

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
