## [Decision Letter · Decision Letter 0]

PONE-D-25-14841The Healthy Brain 9 (HB9): A New Instrument to Characterize Subjective Cognitive Decline, and Detect Anosognosia in Mild Cognitive ImpairmentPLOS ONE

Dear Dr. Galvin,

Thank you for submitting your manuscript to PLOS ONE. After careful consideration, we feel that it has merit but does not fully meet PLOS ONE’s publication criteria as it currently stands. Therefore, we invite you to submit a revised version of the manuscript that addresses the points raised during the review process.

We look forward to receiving your revised manuscript.

Kind regards,

Carlos Tomaz, Ph.D.

Academic Editor

PLOS ONE

Journal Requirements:

Dr Galvin is the creator of the HB9 and the copyright is held by the University of Miami Miller School of Medicine. Dr Galvin is Chief Scientific Officer for Cognivue, Inc and receives consulting fees. The other authors have nothing to disclose. The authors take full responsibility for the data and have the right to publish all data.

5. Please provide a complete Data Availability Statement in the submission form, ensuring you include all necessary access information or a reason for why you are unable to make your data freely accessible. If your research concerns only data provided within your submission, please write "All data are in the manuscript and/or supporting information files" as your Data Availability Statement.

6. Please remove all personal information, ensure that the data shared are in accordance with participant consent, and re-upload a fully anonymized data set.

Reviewers' comments:

Reviewer's Responses to Questions

**Comments to the Author**

1. Is the manuscript technically sound, and do the data support the conclusions?

Reviewer #1: Yes

2. Has the statistical analysis been performed appropriately and rigorously? 

Reviewer #1: No

3. Have the authors made all data underlying the findings in their manuscript fully available?

Reviewer #1: No

4. Is the manuscript presented in an intelligible fashion and written in standard English?

Reviewer #1: Yes

5. Review Comments to the Author

Reviewer #1: Thank you for the opportunity to review this manuscript. It tackles the important and clinically relevant areas of subjective cognitive decline (SCD) and anosognosia in mild cognitive impairment (MCI) by introducing the novel Healthy Brain 9 (HB9) instrument. This is certainly an original contribution, and the initial findings show promise. However, I believe significant revisions are needed to fully realize the potential of this work, particularly regarding the methodology and the interpretation of the results.

Looking at the Introduction, the research question is well-defined and addresses a clear gap in the field. The literature review is reasonably comprehensive and current. To make the manuscript's foundation even stronger, I'd suggest expanding the background discussion to more explicitly compare the HB9 with other existing, validated tools for SCD and anosognosia. Showing precisely how the HB9 overcomes limitations or fills specific niches left by current instruments would strengthen the rationale. Mentioning relevant reporting standards, like the COSMIN guidelines for patient-reported outcomes, early on might also help frame the psychometric approach more effectively.

Turning to the Methodology, the study utilizes a sample size and diversity that appear adequate for good external validity, and the documentation of recruitment and ethical procedures (IRB approval, consent) is thorough. The choice of statistical analyses like Cronbach's alpha, ROC curves, regression, and ANOVA is generally appropriate for the initial validation goals. However, a key limitation is the cross-sectional design. While suitable for a first look, it inherently prevents drawing conclusions about causality or the progression of SCD/anosognosia over time. This should be clearly acknowledged, perhaps with stronger recommendations for future longitudinal studies. Another point needing careful attention is the potential for recall bias due to the 5-year retrospective self-report period. While acknowledged, the manuscript could delve deeper into this limitation and ideally suggest concrete ways to mitigate it in future research, such as using ecological momentary assessment (EMA). There's also a critical discrepancy concerning the QDRS cut-off score – the text mentions ≥2, while Table 1 seems to use a mean of 1.6. This inconsistency needs immediate clarification and correction to ensure the methodology is reproducible and sound. Finally, while the statistical methods chosen are standard, the reported moderate sensitivity in the ROC analyses raises questions about the HB9's clinical utility for detecting subtle cognitive changes. It might be beneficial to explore more advanced machine learning techniques (like random forests or gradient boosting) with cross-validation, which could potentially improve diagnostic accuracy. Using specialized software like R (with packages such as "mirt" or "lavaan") or Python could also add a layer of analytical rigor and reproducibility.

In the Results section, the findings are generally presented clearly, and the tables are helpful for understanding the psychometric properties, group comparisons, and ROC analyses. However, the interpretation could go further by explicitly discussing the clinical significance of the findings, especially the implications of the moderate sensitivity observed in the ROC curves. What does this level of sensitivity mean practically for clinicians? The relatively weak correlations found between HB9 scores and objective cognitive tests or biomarkers also warrant more critical examination. What might this suggest about the construct the HB9 is measuring? Is it capturing subjective concerns distinct from objective performance? Furthermore, the influence of education seems somewhat underexplored. Simply noting a modest correlation isn't quite enough; conducting subgroup analyses or using methods like structural equation modeling (SEM) to probe how education moderates HB9 scores would provide valuable insights.

The Discussion does a good job of placing the findings within the context of existing literature and outlining potential clinical implications. The authors also effectively acknowledge limitations like the cross-sectional design and educational skew in the sample. However, the discussion would be strengthened by a deeper exploration of the potential impact of recall bias and the practical consequences of the moderate diagnostic sensitivity. When recommending future research, being more explicit about proposing longitudinal designs, EMA, advanced statistical methods, and perhaps psychometric refinements using Item Response Theory (IRT) or Confirmatory Factor Analysis (CFA) would provide a clearer roadmap. Referencing how the validation efforts align, or could better align, with standards like COSMIN would also add weight to the psychometric discussion.

Overall, the manuscript is logically structured, and the figures and tables are clear. Adherence to ethical standards seems good, although adding a specific statement about potential conflicts of interest or intellectual property related to the HB9 instrument itself would enhance transparency.

6. PLOS authors have the option to publish the peer review history of their article (what does this mean? ). If published, this will include your full peer review and any attached files.

**Do you want your identity to be public for this peer review?** For information about this choice, including consent withdrawal, please see our Privacy Policy .

Reviewer #1: **Yes: ** Danilo Assis Pereira

---

## [Author Response · Author response to Decision Letter 1]

7 May 2025

Response to Review

PONE-D-25-14841

Corresponding Author: James E. Galvin, MD, MPH

We thank the reviewers for the careful and thoughtful review of the manuscript. We carefully considered each comment and addressed them in the revised manuscript. Changes are highlighted and exact changes are detailed by section and line numbers for easy identification.

Academic Editor

1. Please ensure that your manuscript meets PLOS ONE's style requirements, including those for file naming

AUTHOR RESPONSE: This has been completed as requested

2. We note that the grant information you provided in the ‘Funding Information’ and ‘Financial Disclosure’ sections do not match

AUTHOR RESPONSE: This has been completed as requested

3. Please confirm that this does not alter your adherence to all PLOS ONE policies on sharing data and materials, by including the following statement: "This does not alter our adherence to PLOS ONE policies on sharing data and materials.”

AUTHOR RESPONSE: This has been completed as requested

4. Your ethics statement should only appear in the Methods section of your manuscript.

AUTHOR RESPONSE: This has been completed as requested

5. Please provide a complete Data Availability Statement in the submission form, ensuring you include all necessary access information or a reason for why you are unable to make your data freely accessible.

AUTHOR RESPONSE: This has been completed as requested.

6. Re-upload a fully anonymized dataset.

AUTHOR RESPONSE: This has been completed as requested. All relevant data are available in the Open Science Framework repository: https://doi.org/10.17605/OSF.IO/T9Q8S.

7. Please include captions for your Supporting Information files at the end of your manuscript

AUTHOR RESPONSE: We do not have any supporting information files

8. Please review your reference list to ensure that it is complete and correct.

AUTHOR RESPONSE: This has been completed as requested.

Reviewer #1

1. Thank you for the opportunity to review this manuscript. It tackles the important and clinically relevant areas of subjective cognitive decline (SCD) and anosognosia in mild cognitive impairment (MCI) by introducing the novel Healthy Brain 9 (HB9) instrument. This is certainly an original contribution, and the initial findings show promise. However, I believe significant revisions are needed to fully realize the potential of this work, particularly regarding the methodology and the interpretation of the results.

AUTHOR RESPONSE: We thank the reviewer for recognizing the novelty and importance of the work and for the thoughtful comments on how to improve the manuscript.

2. Looking at the Introduction, the research question is well-defined and addresses a clear gap in the field. The literature review is reasonably comprehensive and current.

AUTHOR RESPONSE: We thank the reviewer for this kind comment.

3. To make the manuscript's foundation even stronger, I'd suggest expanding the background discussion to more explicitly compare the HB9 with other existing, validated tools for SCD and anosognosia. Showing precisely how the HB9 overcomes limitations or fills specific niches left by current instruments would strengthen the rationale. Mentioning relevant reporting standards, like the COSMIN guidelines for patient-reported outcomes, early on might also help frame the psychometric approach more effectively.

AUTHOR RESPONSE: This is an excellent suggestion. Actually, as we reviewed the literature for this manuscript, we could find little evidence that any SCD tool had been extensively examined in MCI/dementia participants or that any anosognosia measurement had been tried in controls with and without cognitive complaints. This gap can be met with the HB9 which can be used in older adults with and without objective cognitive findings to determine what level of subjective complaints they may have. We expanded this in the revised manuscript as follows:

Introduction, lines 126-129: Further, as current instruments were developed to specifically characterize either SCD or anosognosia, there is no literature to suggest how an SCD instrument would perform in MCI with anosognosia, or how an instrument designed to quantify anosognosia in dementia would capture SCD

Discussion, lines 462-467: There is little literature to suggest how instruments designed for SCD would work in MCI with and without anosognosia and even less information about how an anosognosia instrument could be used in cognitively normal controls. While the E-Cog has been examined in both SCD and in the calculation of the Awareness of Cognitive Decline measurement, this required information from both the patient and a knowledgeable informant which limits its utility in the clinical setting where study partners rarely accompany patients.

Discussion lines 488-496: Strengths of the HB9 include the ability to use a single instrument to detect SCD in cognitively normal controls and anosognosia in cognitively impaired individuals. The measurement properties of the HB9 align with many components of the consensus-based standards for the selection of health measurement instruments COSMIN taxonomy [66] with 6 of 9 properties established (internal consistency, face validity, criterion validity, structural validity, cross-cultural validity, and hypothesis testing). Future research in the HB9 should focus on establishing intra-rater reliability and measurement error (testing the same rater with repeated administration) and responsiveness (measuring change in outcomes over time).

4. Turning to the Methodology, the study utilizes a sample size and diversity that appear adequate for good external validity, and the documentation of recruitment and ethical procedures (IRB approval, consent) is thorough. The choice of statistical analyses like Cronbach's alpha, ROC curves, regression, and ANOVA is generally appropriate for the initial validation goals. However, a key limitation is the cross-sectional design. While suitable for a first look, it inherently prevents drawing conclusions about causality or the progression of SCD/anosognosia over time. This should be clearly acknowledged, perhaps with stronger recommendations for future longitudinal studies.

AUTHOR RESPONSE: Another excellent suggestion. We strengthened the discussion by including the need for longitudinal studies on the HB9 – which is ongoing in our center.

Discussion, lines 495-496: Longitudinal studies will be needed to draw conclusions about causality and the progression of SCD or anosognosia over time.

5. Another point needing careful attention is the potential for recall bias due to the 5-year retrospective self-report period. While acknowledged, the manuscript could delve deeper into this limitation and ideally suggest concrete ways to mitigate it in future research, such as using ecological momentary assessment (EMA).

AUTHOR RESPONSE: This is a valid point. Although many patient-reported outcomes require the recall of past information for self-reporting (including of the existing instruments for SCD and anosognosia), there may be inherent self-recall biases. We acknowledge this in the discussion in the revised manuscript.

Discussion, lines 482-485: There is the potential for recall bias using the 5-year retrospective self-report period. This is inherent in any instrument requiring past reporting common to nearly all patient-reported outcomes. Future studies could include methods to mitigate recall bias such as ecological momentary assessment.

6. There's also a critical discrepancy concerning the QDRS cut-off score – the text mentions ≥2, while Table 1 seems to use a mean of 1.6. This inconsistency needs immediate clarification and correction to ensure the methodology is reproducible and sound.

AUTHOR RESPONSE: We thank the reviewers for pointing this out. We left out a detail in the original manuscript that permits the research clinician to use the memory domain in the QDRS in addition to the total score to assess whether there is a subjective complaint. This explains the mean scores for SCD vs NCI. For the MCI group, the presence of anosognosia explains the QDRS means. We corrected this in the revised manuscript in the methods section:

Methods, lines 231-237: No Cognitive Impairment (NCI), SCD, or MCI. NCI individuals had normal objective cognitive performance, offered no subjective complaints during the semi-structured CDR interview, had QDRS scores <1.5 or a score of 0 in the memory domain, and were rated CDR 0 and GDS 1. SCD individuals had normal objective cognitive performance, endorsed subjective complaints that their cognition was declining during the semi-structured CDR interview, had scores 2 or greater on the QDRS or a score >0 in the memory domain, and were rated CDR 0 and GDS 2.

7. Finally, while the statistical methods chosen are standard, the reported moderate sensitivity in the ROC analyses raises questions about the HB9's clinical utility for detecting subtle cognitive changes. It might be beneficial to explore more advanced machine learning techniques (like random forests or gradient boosting) with cross-validation, which could potentially improve diagnostic accuracy.

AUTHOR RESPONSE: Thank you for this suggestion. While we agree that additional prediction models would improve diagnostic accuracy, the ROC analyses described in this paper are presented to describe the HB9's association with diagnostic status and not to produce a model or tool for diagnosis. With our current sample size, generating models for diagnosis would be inappropriate. Future study will explore the ability of the HB9 and other assessments to predict diagnosis using machine learning models, including random forests, gradient boosting, and multi-model voting ensembles. This will be easier as we consider the longitudinal data we are now collecting. As an aside, we did run a preliminary analysis to consider this comment. We found very little differences from the logistic regression using a random forest approach. Based on these findings, we do not believe that the paper would benefit from including such an analysis.

8. Using specialized software like R (with packages such as "mirt" or "lavaan") or Python could also add a layer of analytical rigor and reproducibility.

AUTHOR RESPONSE: For this study, IBM's SPSS, a specialized statistical software package, was used for all statistical analyses. While R and Python are useful, and many of our investigators (including some of our co-authors) use these for their analyses and publications, the corresponding author is most familiar with SPSS and so this was the software chosen for this publication.

9. In the Results section, the findings are generally presented clearly, and the tables are helpful for understanding the psychometric properties, group comparisons, and ROC analyses. However, the interpretation could go further by explicitly discussing the clinical significance of the findings, especially the implications of the moderate sensitivity observed in the ROC curves. What does this level of sensitivity mean practically for clinicians?

AUTHOR RESPONSE: We agree that further explanation is warranted. As an author, I have generally refrained from over-explaining the results’ significance but fully agree with the reviewer that additional clarification of the significance of the findings and why the specificity and diagnostic odds ratio are more meaningful than the sensitivity is needed. In the revised manuscript, we added the following text:

Results, lines 380-388: While the sensitivity of the HB9 was less than desirable, there was good specificity for discriminating NCI from SCD and for detecting anosognosia in MCI. The sensitivity of any instrument defines the ability to capture true positives while the specificity defines the ability to identify true negatives. For the HB9, specificity is the more useful property since for SCD since we are trying to eliminate the NCI individuals. Similarly, for the study of anosognosia, we are trying to eliminate MCI with awareness. A better way of capturing this clinically is to use diagnostic odds ratios based on the positive and negative likelihood ratios. The diagnostic odds ratio to capture any subjective complaints was 11.2. The diagnostic odds ratio to detect SCD was 15.5, while the diagnostic odds ratio to detect Anosognosia in MCI was 14.4.

10. The relatively weak correlations found between HB9 scores and objective cognitive tests or biomarkers also warrant more critical examination. What might this suggest about the construct the HB9 is measuring? Is it capturing subjective concerns distinct from objective performance?

AUTHOR RESPONSE: This is an interesting point. We believe the HB9 is different from other cognitive measures because it assesses the participants’ point of view of whether their memory, thinking, and everyday activities have changed and how significant that change is impacting their lives. This is quite different from objective performance on a neuropsychological test of a specific domain such as memory, language, or executive function compared to an age- and education-norm. This is played out as to why the HB9 may distinguish NCI from SCD while the neuropsychological tests cannot. Similarly for the HB9 ability to discriminate MCI with awareness from MCI with anosognosia when neuropsychological tests cannot. The same is true for the biomarkers which for most part are AD-specific. We did look at subgroups and found some evidence that the HB9 scores may help identify MCI with AD changes and that NCI individuals have the healthiest brains. These points are expanded upon in the revised manuscript:

Results, lines 328-334: We further explored the biomarkers with subgroup analyses. For the NCI group, higher HB9 scores were correlated with higher A� 42 levels (r=0.239, p=0.009) and higher NFL (r=0.263, p=0.003). For the SCD group, higher HB9 scores were marginally correlated with smaller medial temporal lobe (r=-0.269, p=0.052) and hippocampal (r=-0.251, p=0.070) volumes. For MCI with awareness, there were no significant correlations with any biomarker, while HB9 scores in MCI with anosognosia were strongly correlated with APS2 scores (r=0.609, p<0.001) and ptau217% ratios (r=0.788, p<0.001) suggesting more AD-related neuropathologic changes.

Discussion, lines 411-422: The HB9 does not measure objective performance suggesting that the HB9 captures aspects of intraindividual change noted by the participant, rather than inter-individual change noted by neuropsychological testing. Based on the findings presented, we believe that the HB9 measures the construct of cognition quite differently from traditional objective cognitive measurements because it assesses the participants’ point of view of whether they perceive that their memory, thinking, and everyday activities have changed and if they have changed, how significant that change is impacting their lives. This is fundamentally different from objective performance on a neuropsychological test of a specific domain such as memory, language, or executive function compared to an age- and education-norm. This may explain why the HB9 can distinguish NCI from SCD while the neuropsychological tests cannot. The same may be true for the HB9 ability to discriminate MCI with awareness from MCI with anosognosia when neuropsychological tests cannot.

Discussion, lines 432-434: This is also supported by subgroup analyses where HB9 scores in the NCI group suggested healthier brains and HB9 scores in SCD and MCI with anosognosia were associated with neurodegenerative changes.

11. Furthermore, the influence of education seems somewhat underexplored. Simply noting a modest correlation isn't quite enough; conducting subgroup analyses or using methods like structural equation modeling (SEM) to probe how education moderates HB9 scores would provide valuable insights.

AUTHOR RESPONSE: In response the reviewer’s comments, we conducted subgroup analyses to further explore the educational effect. This is included in the revised manuscript:

Results, lines 310-316: There were educational effects on HB9 scores. Post college graduates (n=133) had lower HB9 scores (3.5+4.0 vs. 5.7+5.4; F=3.3, df 2, 349, p=0.037) compared to individuals with 12 or less years of

---

## [Editor Report · Decision Letter 1]

The Healthy Brain 9 (HB9): A new instrument to characterize subjective cognitive decline, and detect anosognosia in mild cognitive impairment

PONE-D-25-14841R1

Dear Dr. James E. Galvin,

We’re pleased to inform you that your manuscript has been judged scientifically suitable for publication and will be formally accepted for publication once it meets all outstanding technical requirements.

Kind regards,

Carlos Tomaz, Ph.D.

Academic Editor

PLOS ONE

---

## [Editor Report · Acceptance letter]

PONE-D-25-14841R1

PLOS ONE

Dear Dr. Galvin,

I'm pleased to inform you that your manuscript has been deemed suitable for publication in PLOS ONE. Congratulations! Your manuscript is now being handed over to our production team.

Kind regards,

on behalf of

Dr. Carlos Tomaz

Academic Editor

PLOS ONE